# Incidence of acute lower respiratory tract disease hospitalisations, including pneumonia, among adults in Bristol, UK, 2019, estimated using both a prospective and retrospective methodology

Catherine Hyams [1,2] Elizabeth Begier [3] Maria Garcia Gonzalez,[4,5] Jo Southern,[6] James Campling,[6] Sharon Gray,[3] Jennifer Oliver,[2,7] Bradford D Gessner [8] Adam Finn[2,4]

For numbered affiliations see end of article.

**Correspondence to**
Dr Catherine Hyams;
catherine.hyams@bristol.ac.uk

## ABSTRACT

**Objectives** To determine the disease burden of acute lower respiratory tract disease (aLRTD) and its subsets (pneumonia, lower respiratory tract infection (LRTI) and heart failure) in hospitalised adults in Bristol, UK.

**Setting** Single-centre, secondary care hospital, Bristol, UK.

**Design** We estimated aLRTD hospitalisations incidence in adults (≥18 years) in Bristol, UK, using two approaches. First, retrospective International Classification of Diseases 10th revision (ICD-10) code analysis (first five positions/hospitalisation) identified aLRTD events over a 12-month period (March 2018 to February 2019). Second, during a 21-day prospective review (19 August 2019 to 9 September 2019), aLRTD admissions were identified, categorised by diagnosis and subsequently annualised. Hospital catchment denominators were calculated using linked general practice and hospitalisation data, with each practice's denominator contribution calculated based on practice population and per cent of the practices' hospitalisations admitted to the study hospital.

**Participants** Prospective review: 1322 adults screened; 410 identified with aLRTD. Retrospective review: 7727 adult admissions.

**Primary and secondary outcome measures** The incidence of aLRTD and its subsets in the adult population of Southmead Hospital, Bristol UK.

**Results** Based on ICD-10 code analysis, annual incidences per 100 000 population were: aLRTD, 1901; pneumonia, 591; LRTI, 739; heart failure, 402. aLRTD incidence was highest among those ≥65 years: 65–74 (3684 per 100 000 adults), 75–84 (6962 per 100 000 adults) and ≥85 (11 430 per 100 000 adults). During the prospective review, 410/1322 (31%) hospitalised adults had aLRTD signs/symptoms and annualised incidences closely replicated retrospective analysis results.

**Conclusions** The aLRTD disease burden was high, increasing sharply with age. The aLRTD incidence is probably higher than estimated previously due to criteria specifying respiratory-specific symptoms or radiological

## STRENGTHS AND LIMITATIONS OF THIS STUDY

⇒ We used two analytical methods at the same site over a comparable period, to calculate incidence using both prospective and retrospective approaches.

⇒ The case burden of acute lower respiratory tract disease (aLRTD) and its subgroups was predefined and included patients with atypical presentations.

⇒ We calculated incidence using a denominator derived from general practitioner records, providing increased accuracy compared with population calculations based on census data.

⇒ This was a single-centre study, with a predominantly Caucasian cohort; therefore, the findings might not be generalisable to other populations.

⇒ The International Classification of Diseases 10th revision coding data analysis was limited to codes within the first five positions, and therefore may have excluded some cases where other diagnoses were placed higher in the diagnostic coding hierarchy.

change, usage of only the first diagnosis code and mismatch between case count sources and population denominators. This may have significant consequences for healthcare planning, including usage of current and future vaccinations against respiratory infection.

## INTRODUCTION

Acute lower respiratory tract disease (aLRTD) encompasses pneumonia, non-pneumonic lower respiratory tract infection (NP-LRTI), acute bronchitis, exacerbation of underlying respiratory diseases (including asthma and chronic obstructive pulmonary disease (COPD)) and acute heart failure (HF) events resulting in respiratory symptoms (eg, breathlessness). Before

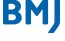

the COVID-19 pandemic, European healthcare costs for pneumonia alone in adults were estimated at €10 billion annually, including €5.7 billion for inpatient care.[1] Pneumonia incidence in Europe varies by country and intracountry region, age, socioeconomic status and gender[2–4]; however, in all studies pneumonia incidence in adults increases sharply with age.[3] Pneumonia affects an estimated 0.5%–1% of UK adults each year.[5 6] Overall LRTI incidence is considerably higher with ~15% of UK adults aged ≥65 years experiencing an event each year.[7] While HF is not typically clinically included as an acute respiratory illness, HF with respiratory symptoms may be caused by respiratory viral infection, such as respiratory syncytial virus (RSV), either acutely or 3–4 weeks after the primary infection.[8 9]

However, aLRTD incidence may be considerably higher than previously reported, given that published literature has documented several reasons why previous estimates may have been erroneously low.[1] Estimates of aLRTD based on pneumonia defined as radiologically demonstrated alveolar infiltrates, may underestimate true disease burden as chest radiography (CXR) is an imperfect gold-standard.[10 11] Immunosuppressed, elderly or dehydrated patients are likely to be under-represented if respiratory infection is defined by radiologically demonstrated changes.[10 11] Microbiological investigations for pneumonia are undertaken variably and identify a causative pathogen in 50% of cases at most[12 13]; hence, the disease is probably under-reported when confirmed microbiological diagnosis is required. Furthermore, RSV infection has recently been recognised as an important respiratory pathogen later in life,[9] with severe disease occurring in patient groups in whom the diagnosis is likely to be under-recognised (eg, the elderly or those with underlying cardiac conditions).[8] Studies of clinical coding data are retrospective and subject to recognised limitations associated with this methodology.[14 15] Older patients with pneumonia often have atypical presenting signs and symptoms, which may lead to missed or incorrect admission diagnoses.[16] Pneumonia may occur secondary to, or be an underlying cause of, the main presenting report, particularly in patients with cerebrovascular accidents, HF, COPD exacerbations or altered consciousness levels.[17] In these scenarios, pneumonia may not be the primary hospitalisation diagnosis code and may not even be coded as an associated diagnosis.

There are many studies examining the incidence of acute respiratory illness in children; however, data on respiratory illness in adults in the UK is lacking. Given the paucity of data supporting accurate aLRTD incidence rates and its disease subsets in adults, we undertook to assess aLRTD incidence by two approaches (retrospective and prospective) in Bristol, UK, seeking to determine the disease burden of hospitalised aLRTD and its subgroups more accurately.

## METHODS
### Study design
This study was conducted at a large secondary care institution in UK (North Bristol NHS Trust) with specialist respiratory services (interstitial lung disease, pleural disease). Two approaches were undertaken to estimate aLRTD incidence: (1) 'retrospective analysis' of aLRTD International Classification of Diseases 10th revision (ICD-10) diagnostic codes for an entire year; and, (2) 21-day observational 'prospective review' of aLRTD hospital admissions.

### Patient and public involvement
No patient involved.

### Retrospective analysis
For the retrospective analysis, all adult inpatient admissions (≥18 years) obtained from Hospital Episode Statistic to the study hospital during March 2018 to February 2019 with aLRTD ICD-10 diagnostic codes (online supplemental data 1) in any of the first five positions were identified and categorised into aLRTD subgroups: pneumonia, NP-LRTI, other lower respiratory tract disease (LRTD) and HF (online supplemental data 2). A mutually exclusive hierarchy was used (pneumonia, NP-LRTI, then other LRTD) although HF diagnoses could co-occur with other categories. 'Other LRTD' included acute respiratory events that could not definitively be placed in another category. Only the first five ICD-10 codes were available for analysis.

### Prospective review
Adult patients (≥18 years) resident within Bristol, North Somerset and South Gloucestershire Clinical Commissioning Group (CCG) referred to the acute medical unit (AMU) at North Bristol NHS Trust during 19 August 2019 to 9 September 2019 were included in an audit on acute respiratory illness. This time period was selected because it was felt to represent a period when there were an average number of adults hospitalised with aLRTD. A respiratory physician (CH) reviewed presenting features and investigation results for each admitted patient to determine whether aLRTD was present. Further medical record review was undertaken if patients had: new/worsening breathlessness, cough or sputum production; new or worsening peripheral oedema; pleurisy; clinical examination findings consistent with respiratory infection or HF; or, fever attributable to suspected respiratory infection. Patients with non-respiratory diagnoses were excluded. There were no patient refusals for either approach.

*Prospective Review Outcome measures*

aLRTD was considered confirmed in individuals with: new/worsening respiratory symptoms, with or without fever; sepsis, delirium or raised inflammatory markers attributable to likely respiratory infection in admitting clinical team's opinion; radiological change in keeping with infection (eg, consolidation); and/or final diagnosis of NP-LRTI, pneumonia or infective exacerbation of a chronic respiratory condition. A pneumonia diagnosis

was assigned if radiological changes likely due to infection were described by the reporting radiologist. An NP-LRTI diagnosis was assigned if aLRTD signs and symptoms likely to be due to infection were present without demonstrated radiological change. An HF diagnosis was assigned in presence of: new/worsening breathlessness and bi-basal crepitations, cardiac wheeze or new/worsening bilateral pitting oedema; elevated pro-NT BNP (N-terminal pro B-type natriuretic peptide) (≥450 pg/mL); radiologist-reported radiographic changes consistent with cardiac failure; or a final consultant physician diagnosis of HF, cardiac failure or left-ventricular failure. When present, ≥1 diagnosis was selected.

For both retrospective and prospective studies, pneumonia included both community and healthcare setting acquired cases; although, the prospective review only captured admitting diagnoses and pneumonias occurring later during hospitalisation were not included.

### Incidence calculations

Annual incidence per 100 000 persons was calculated for both retrospective and prospective studies. Case counts from prospective review were annualised (ie, case counts by diagnosis and overall were divided by the percentage of annual admissions for these diagnosis groups that occurred in this 21-day period in the retrospective analysis).

### Incidence denominators

To calculate appropriate population denominators for incidence calculations, aLRTD hospital admission event data were linked to aggregated general practitioner (GP) practice patient registration data within the NHS Bristol, North Somerset and South Gloucestershire CCG for 2017–2019. Only 4% of patients sought care at North Bristol NHS Trust hospital from outside these local CCGs,

despite presence of specialist respiratory services. In the UK, GP registration is available free of charge for all, regardless of residential status. For GP practices within these same CCGs, the proportion of their aLRTD admissions occurring at North Bristol NHS Trust was multiplied by their patient registration count in 2019 by age group, to get each practice's contribution to the denominator (eg, if 50% aLRTD admissions were at North Bristol among persons 50–64 years, the practice would contribute half of their patients 50–64 years to the denominator). Further details of this methodology have been described previously.[18]

### Statistical analysis

Patient characteristics were tabulated by aLRTD diagnosis. Categorical variables were presented as counts with percentages. Continuous data are presented with means and SD if normally distributed and medians and IQR if not normally distributed. Patient groups difference were evaluated using the Friedman test with Wilcoxon signed-rank test.

## RESULTS
### Retrospective analysis

Over a 12-month period, we identified 7727 hospital admissions for aLRTD: 3005 NP-LRTI admissions, 2402 pneumonia, 1633 HF and 1071 other LRTD (table 1). The aLRTD admissions were lowest in March and April and highest December through February (figure 1A), overall and for all aLRTD subgroups (p<0.05) (figure 1B–D). Overall, 28.1% (2244) cases were identified as being potentially hospital-acquired infection based on co-occurring ICD-10 code 'Y95 Nosocomial Infection'.

**Table 1** Demographic characteristics of patients admitted with acute lower respiratory tract disease for 1-year International Classification of Diseases 10th revision code retrospective analysis and 21-day prospective review period—2018–2019

| Characteristic | Pneumonia | | NP-LRTI | | Heart failure | | Other LRTD | All LRTD | |
|---|---|---|---|---|---|---|---|---|---|
| Study | Prospective review | Retrospective analysis | Prospective review | Retrospective analysis | Prospective review | Retrospective analysis | Retrospective review only | Prospective review | Retrospective analysis |
| N | 152 | 2402 | 188 | 3005 | 77 | 1633 | 1071 | 410 | 7727 |
| Gender, females | 61 (40) | 1078 (45) | 99 (53) | 1482 (49) | 39 (51) | 731 (45) | 489 (46) | 194 (47) | 3780 (49) |
| Age | | | | | | | | | |
| Median (IQR), years | 80 (67–86) | 81 (66–88) | 70 (46–87) | 69 (45–87) | 87 (72–90) | 87 (70–90) | 74 (53–82) | 80 (64–88) | 81 (65–90) |
| 18–24 | 4 (3) | 72 (3) | 9 (5) | 151 (5) | 0 (0) | 0 (0) | 26 (2) | 13 (3) | 249 (3) |
| 25–34 | 2 (2) | 48 (2) | 12 (6) | 183 (6) | 0 (0) | 3 (0) | 33 (3) | 14 (3) | 267 (3) |
| 35–44 | 6 (4) | 97 (4) | 14 (7) | 209 (7) | 2 (3) | 10 (1) | 59 (6) | 22 (5) | 375 (5) |
| 45–54 | 8 (5) | 118 (5) | 11 (6) | 183 (6) | 0 (0) | 22 (1) | 112 (10) | 19 (5) | 435 (6) |
| 55–64 | 18 (12) | 293 (12) | 19 (10) | 305 (10) | 8 (10) | 158 (10) | 210 (20) | 45 (11) | 966 (13) |
| 65–74 | 34 (22) | 501 (21) | 32 (17) | 549 (18) | 10 (15) | 199 (12) | 223 (21) | 75 (18) | 1472 (19) |
| 75–84 | 44 (28) | 667 (28) | 40 (21) | 621 (21) | 20 (30) | 498 (31) | 205 (19) | 100 (24) | 1991 (26) |
| ≥85 | 38 (26) | 606 (25) | 51 (27) | 704 (23) | 37 (55) | 742 (45) | 203 (19) | 123 (30) | 2255 (29) |

LRTD, lower respiratory tract disease ; NP-LRTI, non-pneumonic lower respiratory tract infection .

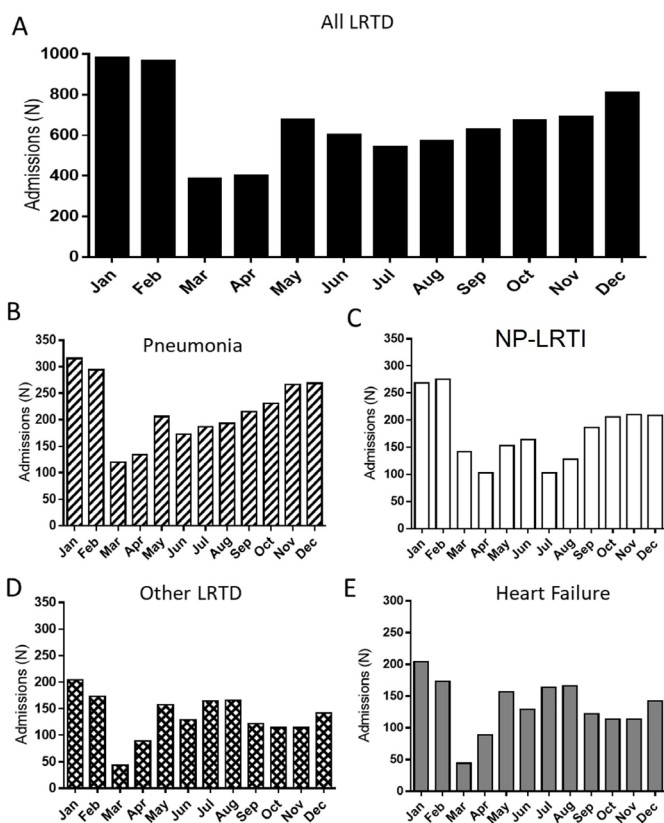

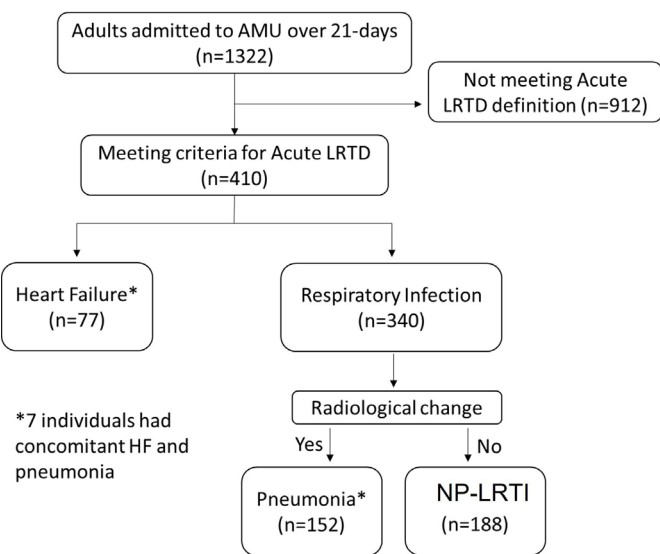

**Figure 1** The aLRTD admissions identified by retrospective International Classification of Diseases 10th revision (ICD-10) diagnostic code analysis at North Bristol National Health Service Trust—UK 2018–2019. Monthly number of patients admitted, based on ICD-10 coding analysis, with (A) all acute lower respiratory tract disease (aLRTD) (black bars), (B) pneumonia (slashed bars), (C) non-pneumonic lower respiratory tract infection (NP-LRTI) (white bars), (D) other LRTD (cross-hash bars) and (E) heart failure (grey bars).

**Figure 2** Flow diagram of the prospective review. AMU, acute medical unit; HF, heart failure; LRTD, lower respiratory tract disease; NP-LRTI, non-pneumonic lower respiratory tract infection.

### Prospective review
Among 1322 eligible adult patients referred to AMU over the 21-day review period (figure 2), 410 patients had signs or symptoms of aLRTD: 188 (46%) NP-LRTI, 152 (37%) pneumonia and 77 (19%) HF. Seven patients had both decompensated HF and a respiratory infection at hospital admission. On admission, >10% of patients with aLRTD did not have respiratory symptoms: 16 (11%) pneumonia, 25 (13%) NP-LRTI and 18 (14%) HF (table 2).

Almost all adults admitted with aLRTD underwent routine biochemistry, haematology and radiological investigation (99.9%, n=409). In contrast, only 150 (37%) patients with aLRTD had microbiological testing performed: blood cultures (n=149, 36%) and urine cultures (n=143, 35%). Pneumonia patients more commonly underwent microbiological investigation than patients with NP-LRTI (p<0.05) with highest disparity in rates of sputum culture, urinary antigens and respiratory viral PCR (table 2). All patients with cardiac failure who underwent microbiological investigation had concomitant respiratory infection (table 2). Overall, a microbiological diagnosis was found in 11 (3%) cases, highlighting

the low frequency of definitive pathogen identification in aLRTD. Younger patients underwent microbiological testing more frequently than the elderly for all aLRTD categories (table 2).

### Disease incidence
Retrospective analysis yielded an overall aLRTD incidence of 1901 per 100 000. Disease incidence rose with increasing age (table 3), both overall and for all disease subgroups; incidences per 100 000 among adults aged ≥85 years were: 11 430 (aLRTD), 6116 (NP-LRTI), 4215 (pneumonia) and 4005 (HF). Overall, 28.1% aLRTD hospitalisations also included an ICD-10 discharge code for 'nosocomial infection', suggesting the aLRTD event or other infection was hospital-acquired. For pneumonia 25.3% had the nosocomial infection code. If all of these were related to the pneumonia diagnosis, an estimated residual 1794 events would have been community-acquired pneumonia (CAP) (annual incidence 441/100 000 adults). Among older age categories, where the incidence of aLRTD was highest, NP-LRTI incidence was similar to pneumonia incidence, with an approximate 1:1 ratio of NP-LRTI to pneumonia cases. However, among adults under age 50 years, there were approximately twice as many NP-LRTI cases observed as pneumonia cases. Incidence calculations using annualised prospective review results were broadly comparable with retrospective analysis of ICD-10 data (table 3).

### DISCUSSION
This is the first UK study to assess aLRTD incidence comprehensively. We conducted an analysis of 12 months of hospital admissions by ICD-10 diagnosis code data and a 21-day prospective review at a large academic hospital in South West England. With both approaches, we found

**Table 2** Clinical characteristics and investigations of patients admitted with acute lower respiratory tract disease over 21-day prospective review period in August to September 2020

| Characteristic | Pneumonia, n=152 (%) | NP-LRTI, n=188 (%) | Heart failure, n=77 (%) | All LRTD, n=410 (%) |
|---|---|---|---|---|
| GP | 56 (37) | 72 (39) | 30 (39) | 158 (39) |
| A&E department | 93 (61) | 100 (54) | 45 (58) | 238 (58) |
| Transfer from another unit | 2 (1) | 13 (7) | 0 (0) | 15 (4) |
| Other | 1 (1) | 1 (1) | 2 (3) | 4 (1) |
| Referral source | | | | |
| Typical features* | 136 (89) | 163 (87) | 63 (82) | 355 (87) |
| Atypical features | 16 (11) | 25 (13) | 14 (18) | 55 (13) |
| Collapse/falls | 11 (7) | 12 (6) | 0 (0) | 23 (6) |
| Confusion | 0 (0) | 7 (4) | 4 (5) | 10 (2) |
| Drowsiness | 1 (1) | 1 (1) | 2 (3) | 4 (1) |
| Off legs/generally unwell | 5 (3) | 5 (3) | 8 (10) | 18 (4) |
| LRTD signs and symptoms on referral to AMU | | | | |
| Biochemistry | 152 (100) | 185 (99) | 77 (100) | 419 (100) |
| Haematology | 152 (100) | 185 (99) | 77 (100) | 419 (100) |
| Radiology | 152 (100) | 185 (99) | 77 (100) | 419 (100) |
| Investigations performed | | | | |
| Testing by age group | | | | |
| All patients | 79/152 (52)† | 77/188 (41) | 11/77 (14) | 167 (41) |
| 18–24 | 3/4 (75) | 6/9 (66) | 0/0 (0) | 9/13 (69) |
| 25–34 | 0/0 (0) | 6/12 (50) | 0/0 (0) | 6/12 (50) |
| 35–44 | 5/6 (83) | 10/14 (71) | 2/2 (100) | 17/22 (77) |
| 45–54 | 6/8 (75) | 6/11 (55) | 0/0 (0) | 13/19 (68) |
| 55–64 | 11/18 (61) | 12/19 (63) | 5/8 (63) | 31/45 (69) |
| 65–74 | 15/34 (44) | 12/32 (38) | 1/10 (10) | 28/75 (37) |
| 75–84 | 21/43 (49) | 10/40 (25) | 2/20 (10) | 33/100 (33) |
| ≥85 | 18/39 (46) | 15/51 (19) | 1/37 (3) | 34/124 (27) |
| Test performed | | | | |
| Blood culture | 79 (52) | 70 (37) | 5 (6) | 150 (37) |
| Urine culture | 66 (43) | 77 (41) | 11 (14) | 150 (37) |
| Sputum culture | 27 (18)† | 7 (4) | 2 (3) | 35 (9) |
| BinaxNOW Pn UAT | 29 (19)† | 6 (3) | 0 (0) | 35 (9) |
| Respiratory virus PCR | 16 (11)† | 11 (6) | 1 (1) | 28 (7) |
| Pleural fluid culture | 3 (2) | 0 (0) | 1 (1) | 4 (1) |

*Typical symptoms included cough, breathlessness, increased or discoloured sputum production, wheeze, pleurisy, peripheral oedema, haemoptysis, reduced exercise tolerance and/or fever.
†P<0.05.
‡BinaxNOW Pn UAT was only performed in accordance with NICE/BTS guidelines.
A&E, accident and emergency department; AMU, acute medical unit; BTS, British Thoracic Society; GP, general practitioner; LRTD, lower respiratory tract disease; NICE, National Institute for Health and Care Excellence; NP-LRTI, non-pneumonic lower respiratory tract infection; Pn UAT, pneumococcal urinary antigen test.

a high annual incidence of aLRTD (>1700 per 100 000; 1.7%), pneumonia (~0.6%), NP-LRTI without pneumonia (>0.7%) and HF (>0.4%). Incidences increased sharply in a non-linear manner as age increased above 65 years for all aLRTD categories. These results suggest rates are probably significantly higher than previous disease estimates from the UK (table 4) but comparable with many results globally,[19][20] with important consequences for healthcare resources. For example, a recent review highlighted that pneumonia incidences ranged from 1000 to 2500 per 100 000 (1%–2.5%) among persons aged 65–74 years in Spain, Germany, France, Japan and the USA, which are comparable to the >1250 per 100 000 (1.3%) reported here. Some of the potential sources of

**Table 3** Incidence of aLRTD resulting in hospital admission based on prospective and retrospective approaches by age group and condition, North Bristol National Health Service Trust—UK 2018–2019

| | Age groups | | | | | |
| --- | --- | --- | --- | --- | --- | --- |
| | **All adults** | **18–49 years** | **50–64 years** | **65–74 years** | **75–84 years** | **≥85 years** |
| Population in 2018 | 406 481 | 226 920 | 91 534 | 45 705 | 29 487 | 12 835 |
| Retrospective analysis of a year's ICD-10 codes | | | | | | |
| Annual cases—N (row %) | | | | | | |
| All aLRTD | 7727 | 1130 (14) | 1103 (14) | 1684 (22) | 2053 (27) | 1757 (23) |
| Pneumonia | 2402 | 264 (11) | 288 (12) | 589 (25) | 720 (30) | 541 (22) |
| NP-LRTI | 3005 | 576 (19) | 410 (14) | 572 (19) | 662 (22) | 785 (26) |
| Other LRTD | 1071 | 246 (23) | 268 (25) | 226 (21) | 200 (19) | 131 (12) |
| Heart failure | 1633 | 48 (3) | 189 (12) | 397 (24) | 485 (30) | 514 (31) |
| NP-LRTI/pneumonia ratio | 1.3 | 2.2 | 1.4 | 1.0 | 0.9 | 1.5 |
| Incidence (per 100 000) | | | | | | |
| All aLRTD | 1901 | 497 | 1205 | 3684 | 6962 | 13 689 |
| Pneumonia | 591 | 116 | 315 | 1289 | 2442 | 4215 |
| NP-LRTI | 739 | 254 | 448 | 1252 | 2245 | 6116 |
| Other LRTD | 263 | 108 | 293 | 494 | 678 | 1021 |
| Heart failure | 402 | 21 | 206 | 869 | 1645 | 4005 |
| 21-day prospective review (annualised) | | | | | | |
| Annualised cases—N (row %) | | | | | | |
| All aLRTD | 7885 | 1038 | 962 | 1692 | 2231 | 1962 |
| Pneumonia | 2621 | 224 | 397 | 776 | 690 | 534 |
| NP-LRTI | 3857 | 796 | 531 | 653 | 1061 | 816 |
| Heart failure | 2000 | 51 | 205 | 308 | 641 | 795 |
| NP-LRTI/pneumonia ratio | 1.5 | 3.6 | 1.3 | 0.8 | 1.5 | 1.5 |
| Incidence (per 100 000) | | | | | | |
| All aLRTD | 1940 | 458 | 1050 | 3703 | 7565 | 15 283 |
| Pneumonia | 645 | 99 | 433 | 1698 | 2339 | 4164 |
| NP-LRTI | 944 | 351 | 580 | 1429 | 3599 | 6360 |
| Heart failure | 492 | 23 | 224 | 673 | 2174 | 6193 |

Pneumonia category includes community and healthcare acquired. For retrospective ICD-10 based cohort, the following mutually exclusive hierarchy was used to define pneumonia, LRTI and other LRTD; heart failure event could overlap with other categories.
'Other LRTD' contains LRTD events that could not definitively be placed in one of the other respiratory disease categories.
aLRTD, acute lower respiratory tract disease ; HF, Heart Failure; ICD-10, International Classification of Diseases 10th revision; LRTD, lower respiratory tract disease; LRTI, lower respiratory tract infection; NP-LRTI, non-pneumonic lower respiratory tract infection ; pro-NT BNP, N-terminal pro B-type natriuretic peptide.

underestimation for other UK incidence studies (table 4) include: case source and denominator mismatch; use of first position in ICD-code analysis only; inclusion of enrolled subjects in consent-based prospective studies only; requiring specific symptoms and chest X-ray confirmation in order to be classified as pneumonia/aLRTD; and, lastly, the rising incidence of aLRTD.

### Comparison with published literature

No studies have reported aLRTD incidence comprehensively in UK hospitalised patients within the last 20 years. However, eight publications report incidence of ≥1 aLRTD subgroup. Seven publications reported CAP incidence (three from Nottingham, UK). For pneumonia, our incidence estimates were three to fourfold higher than other UK inpatient incidence estimates (table 4) but comparable to estimates from other countries.[19 20] Only two UK studies from approximately 20 years ago reported NP-LRTI incidence (one with both CAP and NP-LRTIs; table 4), and only one provided an inpatient estimate.[21] NP-LRTI incidence was approximately twofold lower than that calculated here, taking into account inclusion of CAP and other NP-LRTI in their estimates.[21 22] The one UK study reporting HF incidence had methodological differences (ie, inclusion of outpatients and limiting to initial HF diagnosis) and estimates could not be compared.[23] Close examination of the existing literature methods yielded multiple sources for potential underestimation.

**Table 4** Literature review of acute lower respiratory tract disease incidence in hospitalised adults, UK

| Study | Study years | Location (facility) | Event setting | Age | Case definition* | Key inclusion | Denominator source | Overall incidence | Age breakdown (years) | Incidence per 100 000 by age† | Comments |
|---|---|---|---|---|---|---|---|---|---|---|---|
| **Community-acquired pneumonia** | | | | | | | | | | | |
| Current study | 2018–2019 | Bristol (Southmead Hospital) | Inpatients only | ≥18 years | Clinical signs/symptoms with radiological change in keeping with infection (prospective review portion). AND Retrospective ICD-10 code analysis (first five positions): J12–J18, J85 and J86. | Hospital-acquired pneumonia (HAP) included | Based on number of persons ≥18 years registered in referring GP practices. For practices with split referral patterns, number adjusted for per cent of admissions that came to Southmead. | 648 / 591 | 18–49 / 50–64 / 65–74 / 75–84 / ≥85 | 116 / 315 / 1289 / 2442 / 4215 | Retrospective analysis includes first five positions. |
| Elston 2012, Epidemiol Infect[26] | 2002–2009 | Hull and East Yorkshire Hospitals‡ | Inpatients only | ≥16 years | ICD-10 codes (first position only): J18.0, J18.9, J13X, J18.1 and J15X. | HAP included | Mid-year population estimates for Hull (city) and EroY (Surrounding County) from Office for National Statistics. | 143 (2002) – 207 (2009) | 15–64 / ≥65 | 48.8–84.1 / 543–781 | Fewer ICD-10 codes included than other analyses; Y95 nosocomial infection included. |
| Millet 2013, J Clin Epidemiol[7] | 1997–2011 | UK | Both inpatients and outpatients | ≥65 years | Read and ICD-10 codes; no specified codes provided. For ICD-10, used first diagnosis code for first episode of hospitalisation only. | HAP excluded | Mid-year UK population estimates from Office for National Statistics. | 799 | 65–69 / 70–74 / 75–79 / 80–84 years / 85–89 / ≥90 | 281 / 431 / 694 / 1205 / 2184 / 4194 | Incidence estimates converted to per 100 000 person-years. |
| Pick 2020, Thorax§[25] | 2013–2014 | Nottingham (two large university hospitals) | Inpatients only | ≥16 years | Inclusion criteria: one or more symptom suggestive of LRTI (defined as cough, increasing dyspnoea, sputum production and fever), with evidence of acute infiltrates consistent with respiratory infection on admission radiography, and treated for a diagnosis of CAP. Exclusion criteria: hospitalisation within 10 days of index admission, a diagnosis of tuberculosis or post-obstructive pneumonia. | HAP excluded | Mid-year estimates for the Greater Nottingham area from the Office for National Statistics, including local population data stratified by age group. | 96.3 | 16–49 / 50–64 / 65–74 / 75–84 / ≥85 | 27.3 / 80.2 / 181.3 / 400.6 / 707.5 | Only consented/enrolled subjects included in estimates. Required CXR confirmation but not all LRTI patients had CXR. |
| | 2017–2018 | | | | | | | 158.4 | 16–49 / 50–64 / 65–74 / 75–84 / ≥85 | 29.9 / 146.9 / 310.4 / 559.5 / 1522.6 | Census-derived denominator that may not have fully matched catchment area. Required specific symptoms and evidence of treatment and some CAP events may not have had this information documented. |
| | 2013–2018 | | | | | | | 120.4 | – | – | |

Continued

**Table 4** Continued

| Study | Study years | Location (facility) | Event setting | Age | Case definition* | Key inclusion | Denominator source | Overall incidence | Age breakdown (years) | Incidence per 100 000 by age† | Comments |
|---|---|---|---|---|---|---|---|---|---|---|---|
| Thorrington 2019, BMC Med | 2004–2005 | England | Inpatients only | ≥65 years | ICD-10 codes (first position only): J18 (pneumonia of unspecified causative organism). | HAP included | Mid-year population estimates for England for 2004–2015 from Office for National Statistics. | NA | ≥65 | 829 | Incidence is per 100 000 person-years. Fewer ICD-10 codes included than other analyses. |
| | 2014–2015 | | | | | | | | ≥65 | 1787 | |
| Trotter 2008, EID | 1997–1998 | England | Inpatients only | ≥65 years | ICD-10 codes (first position only): J12–J18. | HAP included | Mid-year population estimates for England for 1997–2004 from the Office for National Statistics. | NA | 65–74 | 263 | Incidence estimates converted to 100 000 population. |
| | | | | | | | | | 75–84 | 684 | |
| | | | | | | | | | ≥85 | 1599 | |
| | 2004–2005 | | | | | | | | 65–74 | 355 | |
| | | | | | | | | | 75–84 | 877 | |
| | | | | | | | | | ≥85 | 2218 | |

**Lower respiratory tract infection**

**Pneumonia**

| Study | Study years | Location (facility) | Event setting | Age | Case definition* | Key inclusion | Denominator source | Overall incidence | Age breakdown (years) | Incidence per 100 000 by age† | Comments |
|---|---|---|---|---|---|---|---|---|---|---|---|
| Current study | 2018–2019 | Bristol (Southmead Hospital) | Inpatients only | ≥18 years | Clinical signs/symptoms of heart failure or elevated pro-NT BNP or radiological change. AND Retrospective ICD-10 code analysis: J09, J10, J11; J20, J21, J22, J40, J41, J42, J44, J45 and J46. | Excludes all pneumonia | Based on number of persons ≥18 years registered in referring GP practices. For practices with split referral patterns, number adjusted for per cent of admissions that came to Southmead. | 802 | 18–49 | 254 | |
| | | | | | | | | | 50–64 | 448 | |
| | | | | | | | | | 65–74 | 1252 | |
| | | | | | | | | 739 | 75–84 | 2442 | |
| | | | | | | | | | ≥85 | 6116 | |
| Lovering 2001, Clinical Microbiol and Infection[21] | 1994–1996 | Bristol (Southmead Hospital) | Inpatients only | ≥16 years | LRTI episodes from four groups by ICD 9/10 codes: (1) CAP; (2) chest infection or acute exacerbation in presence of asthma; (3) chest infection or acute exacerbation in presence of COPD; or (4) bronchitis with no radiological evidence of pneumonia or pre-existing respiratory disease, such as COPD or asthma. No specified codes provided. | Includes community-acquired pneumonia; HAP excluded | No information on denominator provided. | 623 | 16–39 | 151 | Incidence converted to per 100 000 population. Study involved single hospital and no mention of source of denominator mentioned. |
| | | | | | | | | | 40–49 | 175 | |
| | | | | | | | | | 50–59 | 294 | |
| | | | | | | | | | 60–69 | 1086 | |
| | | | | | | | | | 70–79 | 2135 | |
| | | | | | | | | | >79 | 3141 | |

Continued

**Table 4** Continued

| Study | Study years | Location (facility) | Event setting | Age | Case definition* | Key inclusion | Overall incidence | Denominator source | Age breakdown (years) | Incidence per 100 000 by age† | Comments |
|---|---|---|---|---|---|---|---|---|---|---|---|
| Millet 2013, J Clin Epidemiol[7] | 1997–2011 | UK | Both inpatients and outpatients | ≥65 years | Read and ICD-10 codes; no specified codes provided. For ICD-10, used first diagnosis code for inpatient episode only. | Includes community-acquired pneumonia | 12 293 | Mid-year UK population estimates from Office for National Statistics. (Patients were not considered at risk for community-acquired LRTI during an LRTI illness-episode, during an HES hospitalisation, or for 14 days after any HES hospitalisation or CPRD hospital code. This person-time was excluded from denominator.) | 65–69 | 9221 | Incidence converted to per 100 000 person-years. |
| | | | | | | | | | 70–74 | 10 740 | |
| | | | | | | | | | 75–79 | 12 607 | |
| | | | | | | | | | 80–84 | 15 137 | |
| | | | | | | | | | 85–89 | 18 791 | |
| | | | | | | HAP excluded | | | ≥90 | 26 287 | |
| **Heart failure** | | | | | | | | | | | |
| Current study | 2018–2019 | Bristol (Southmead Hospital) | Inpatients only | ≥18 years | Clinical signs/symptoms of heart failure or elevated pro-NT BNP or radiological change. **AND** Retrospective ICD-10 code analysis: I110; I130; I132; I50. | All or first episode — All | 328 | Based on number of persons ≥18 years registered in referring GP practices. For practices with split referral patterns, number adjusted for per cent of admissions that came to Southmead. | 18–49 | 21 | |
| | | | | | | | | | 50–64 | 206 | |
| | | | | | | | | | 65–74 | 869 | |
| | | | | | | | 402 | | 75–84 | 1645 | |
| | | | | | | | | | ≥85 | 4005 | |
| Uijl 2019, Eur J Heart Fail[23] | 2000–2010 | UK | Both inpatients and outpatients | ≥55 years | Four sources of EHR were linked: CPRD primary care records, HES secondary care hospital charges, Myocardial Ischaemia National Audit Project (MINAP) disease registry, and ONS national death registry. HES ICD-10 codes: Heart failure: I110, I130, I132, I260, I50 and I21. Individuals were excluded if they presented a history of HF before their index date in CPRD, HES or MINAP. | First episode at 55 years or older counted | Not Reported | Not reported. | 55–64, M | 360 | Incidence converted to per 100 000 person-years. Included first episode of HF (inpatient or outpatient) at age 55 and up, so repeat episodes not included. |
| | | | | | | | | | 55–64, Female | 190 | |
| | | | | | | | | | 65–74, Male | 1360 | |
| | | | | | | | | | 65–74, Female | 920 | |
| | | | | | | | | | >75, Male | 3440 | |
| | | | | | | | | | >75, Female | 2800 | |

*Add text names for all listed ICD codes in this footnote or appendix.
†For the current study, only age-group estimates from the retrospective analysis were included in this table, but these were closely comparable to the annualised incidence estimates based on the prospective review. See table 3 for full results.
‡Included hospitals were Hull Royal Infirmary, Castle Hill Hospital, Princess Royal Hospital, Scarborough District General Hospital, Bridlington and District Hospital, York Teaching Hospital, Scunthorpe General Hospital, Goole and District Hospital.
§Only the most recent incidence estimates from the Nottingham CAP study were included.
CAP, community-acquired pneumonia; COPD, chronic obstructive pulmonary disease ; CPRD, Clinical Practice Research Datalink; CXR, chest radiography ; EHR, Electronic Health Record; GP, general practitioner; HES, Hospital Episode Statistic ; ICD-10, International Classification of Diseases 10th revision; LRTI, lower respiratory tract infection ; ONS, Office of National Statistics; pro-NT BNP, N-terminal pro B-type natriuretic peptide.

First, for incidence studies that were not countrywide, identifying an appropriate denominator is challenging. Like many other inpatient settings worldwide, UK hospitals' catchment areas for acute treatment are principally driven by geography, but the proportion of any area's residents expected to use the hospital becomes less clear as distance from the hospital increases because catchment areas and populations of different hospitals may overlap. Defining hospital catchment populations based solely on census data cannot account for this variability. Including all geographical areas using the hospital to any extent results in population denominator overestimation and underestimated incidence. Here, we addressed this by calculating population denominators based on hospital utilisation behaviour from referring general practices.

Second, other studies used fewer codes in their ICD code analysis: all limited their analyses to events where the diagnostic code was in the first position (table 4; case definition column), potentially excluding admissions in which pneumonia/NP-LRTI complicated other underlying respiratory diseases, including COPD and asthma. Limiting to first position has been shown to reduce sensitivity for pneumonia events by about 30% (66%–72% sensitive).[22 24] Conversely, the recent British Thoracic Society (BTS) audit on CAP found ~27% of pneumonia events identified by ICD code (J12–18) had no new CXR infiltrates.[6] Even accounting for this potential over coding practice, our estimates remain well above other published UK estimates.

Third, for other prospective studies, exclusion of events where patients did not consent to participation or were not identified by study surveillance processes (often conducted predominately during business hours) can introduce underestimation. Further, other prospective pneumonia studies specifically required documentation of specific symptoms, radiological findings and treatments,[25] potentially excluding those without these features documented in medical records. In our prospective review, approximately 11% did not display typical signs and symptoms of pneumonia and could have been excluded by that requirement. Requiring CXR confirmation has been shown to reduce incidence estimates for pneumonia,[20] although all pneumonia events in our prospective review were radiologically confirmed.

Fourth, trends over time may also contribute to our estimates being higher than previous reports. Our study's estimates are recent, and rising incidence of pneumonia has been documented in all studies that have reported such trends.[25–27]

Finally, this study included, in part, hospital-acquired pneumonias (HAP), which were excluded from estimates calculated in some other studies (table 4). The retrospective analysis may have included more nosocomial infection than the prospective review, as the latter was focused on evaluation of patients at admission for aLRTD and would not have reliably captured events that developed during hospitalisation. 25.3% pneumonia events included a nosocomial infection code, but this code could relate to any nosocomial infection during that hospitalisation.

If all these cases were assumed to be HAP, our estimates CAP incidence would still be well above prior UK estimates: 441/100 000 (≥18 years).

While not impacting all-cause aLRTD incidence estimates discussed above, during the prospective review, we found low rates of microbial investigation which prevented us from generating pathogen-specific incidence estimates. Only 52% of patients with radiologically-confirmed pneumonia underwent microbiological testing during hospitalisation, with even lower rates in other aLRTD subgroups (41% NP-LRTI and 14% HF). Microbiological testing occurred less frequently as age increased, particularly in patients with NP-LRTI. It is possible that, because aLRTD hospitalisations are substantially more common among older persons, less aetiological investigation is performed. Furthermore, clinicians may elect to treat elderly patients with a more pragmatic and less invasive approach. Management guidelines do not require specific pathogen identification to inform treatment choice. Presence only of atypical features on presentation (in this series, 13% NP-LRTI and 11% pneumonia cases) may also reduce the likelihood of timely microbiological testing. Low rates of microbiological testing, and consequently of confirmed microbiological diagnosis, may represent a source of underestimation of pathogen-specific disease incidence in patient groups (ie, testing bias), particularly in elderly patient groups.

### Strengths and limitations of this study

This study has many strengths. First, this study used two analytical methods at the same site over a comparable period, to calculate incidence using both prospective and retrospective approaches. Second, the case burden of aLRTD and its subgroups was pre-defined and included patients with atypical presentations but with clinical and/or radiological diagnoses, who may otherwise have been excluded from analysis. Additionally, we calculated incidence using a denominator derived from GP records, providing increased accuracy compared with population calculations based on census data.

However, the study also had some limitations. This was a single-centre study, with a predominantly Caucasian cohort; therefore, the findings might not be generalisable to other populations both within the UK and in other countries. Different healthcare systems may affect patient treatment preference, and as the National Health Service provides care which is free at the point of access, the hospitalisation rates seen in this study may be different than those in fee or insurance based healthcare systems. Similarly, physician treatment preferences may affect hospitalisation rates, and we have not explored these in this analysis. The ICD-10 coding data analysis was limited to codes within the first five positions, and therefore may have excluded some cases where other diagnoses were placed higher in the diagnostic coding hierarchy. Furthermore, we could not determine how many cases of the 28.1% ICD-10 cases also coded with

nosocomial infection had hospital-acquired respiratory infection rather than other nosocomial infections.

Although the denominator used to calculate incidence was derived from GP records, this was still an estimate as there is no precisely defined denominator hospital catchment. We were unable to exclude patients from outside the local CCGs in the retrospective analysis, due to the way ICD-10 data were obtained. However, these patients were excluded from the prospective review and the incidence calculated was comparable, suggesting any effect that patients attending North Bristol NHS Trust from outside the local CCGs have on incidence estimates is minimal. This may be because any effect of travelling or health-seeking behaviour is bi-directional: while some patients admitted to Southmead hospital were from outside the local area, it is also true that patients with aLRTD within the relevant CCGs would have been admitted to other hospitals. We also acknowledge the 21-day prospective review period was relatively short, not repeated, and may not be fully representative of clinical practice and cases throughout the year. This study was conducted before the emergence of COVID-19, and we think these data will be useful in one of two-ways in the context of COVID-19: (1) either COVID-19 will become endemic, and the data will reflect the first year before a new normal or (2) COVID-19 will abate and it will provide an anchor for understanding incidence during a respiratory viral pandemic.

In conclusion, we found similarly high estimates of LRTD incidence using two different approaches, and these estimates were higher than those obtained previously in the UK. Determining if there is a real increase in incidence, or if this estimate is larger due to more accurate methodology including a more accurate denominator will require ongoing comprehensive surveillance. Nonetheless, combining all types of LRTD highlights the high burden for this important and potentially life-threatening disease group. Incidence assessments require close assessments of potential areas of under ascertainment, including unidentified or unenrolled cases in prospective studies, reduced positions or number of ICD-10 codes included for retrospective studies, and population denominator mismatch for all study types. Our prospective review findings highlight the need to consider atypical clinical presentations for pneumonia and the lack of routine microbiological investigation in many patients with aLRTD required for pathogen-specific aLRTD incidence calculation. Future research should include a fully prospective assessment of aLRTD incidence with comprehensive diagnostic testing across multiple respiratory seasons, to ensure accurate capture of all aLRTD events, particularly in the elderly. Such research should be undertaken given the high and rising aLRTD burden to enable appropriate healthcare planning and identification of interventions which may reduce disease burden.

**Author affiliations**
[1]Academic Respiratory Unit, University of Bristol, Bristol, UK
[2]Bristol Vaccine Centre, University of Bristol, Bristol, UK
[3]Global Medical and Scientific Affairs, Pfizer Inc, New York City, New York, USA
[4]Population Health Sciences, University of Bristol, Bristol, UK
[5]Academic Respiratory Unit, Southmead Hospital, Bristol, UK
[6]Vaccines Medical Affairs, Pfizer Ltd, Tadworth, UK
[7]Schools of Cellular and Molecular Medicine and Population Health Sciences, University of Bristol, Bristol, UK
[8]Vaccines Medical Development, Scientific and Clinical Affairs, Pfizer Inc, Collegeville, Pennsylvania, USA

**Acknowledgements** We would like to acknowledge the assistance of Qi Yan, PhD (Pfizer) who provided indispensable medical writing and literature review support for this manuscript and Harvey Walsh Health who performed the hospital denominator calculation used here. For the denominator analysis, Hospital Episode Statistics (HES) data were re-used with the permission of NHS Digital via Harvey Walsh Limited.

**Contributors** CH, EB, MGG, JS, BDG and AF generated the research questions and analysis plan. CH, EB, MGG, JS, JC, SG and BDG undertook and contributed to the data analysis. AF oversaw the research and data collection which was undertaken by CH and MGG. All authors contributed to the preparation of the manuscript. CH is the guarantor for this study and accepts full responsibility for the study conduct, had access to the data, and controlled the decision to publish.

**Funding** CH was funded by the National Institute for Health Research (NIHR) (NIHR Academic Clinical Fellowship (ACF-2015-25-002)). The views expressed are those of the author(s) and not necessarily those of the NIHR or the Department of Health and Social Care. The remainder of the study funding was from Pfizer (WI255886-1).

**Competing interests** EB, JS, JC, SG and BDG are full-time employees of Pfizer Vaccines and hold stock or stock options. CH is the Principal Investigator of the Avon CAP study (ISRCTN:17354061) which is an investigator-led University of Bristol study funded by Pfizer and has previously received support from the NIHR in an Academic Clinical Fellowship. AF is a member of the Joint Committee on Vaccination and Immunization (JCVI) and chair of the WHO European Technical Advisory Group of Experts on Immunization (ETAGE) committee. In addition to receiving funding from Pfizer as Chief Investigator of this study, he leads another project investigating transmission of respiratory bacteria in families jointly funded by Pfizer and the Gates Foundation. The other authors have no relevant conflicts of interest to declare.

**Patient and public involvement** Patients and/or the public were not involved in the design, or conduct, or reporting, or dissemination plans of this research.

**Patient consent for publication** Not applicable.

**Ethics approval** This study was approved by North Bristol NHS Trust Research Audit Ethics Committee (CA52218). This work was conducted as part of an audit evaluating the patients admitted to Southmead Hospital with signs and symptoms of respiratory disease. Members of the clinical care team undertook the data collection, and only anonymised data was reviewed by research team members who were not part of the clinical care team.

**Provenance and peer review** Not commissioned; externally peer reviewed.

**Data availability statement** No data are available. No additional data available due to the confidential and sensitive nature of the data in this study.

**ORCID iDs**
Catherine Hyams http://orcid.org/0000-0003-3923-1773
Elizabeth Begier http://orcid.org/0000-0002-1287-5416

Bradford D Gessner http://orcid.org/0000-0002-6216-7194

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
