## [Reviewer comments · BMJ Open]

ARTICLE DETAILS

TITLE (PROVISIONAL)	The incidence of acute lower respiratory tract disease hospitalizations, including pneumonia, among adults in Bristol, UK, 2019, estimated using both a prospective and retrospective methodology
AUTHORS	Hyams, Catherine; Begier, Elizabeth; Garcia Gonzalez, Maria; Southern, Jo; Campling, James; Gray, Sharon; Oliver, Jennifer; Gessner, Bradford; Finn, Adam

VERSION 1 – REVIEW

REVIEWER	Prevots, D. Rebecca National Institutes of Health
REVIEW RETURNED	25-Oct-2021

GENERAL COMMENTS	An extremely thorough and well presented analysis of the rate of Lower Respiratory Tract Infections across adult age groups in Bristol, England. The rates of LRTD were generally similar for both the retrospective and prospective components, which supports the validity of the findings. The low rates of microbiologic sampling are noteworthy, particularly in the older age groups. The strengths and limitations are clearly presented. Overall, this paper presents and informative and important review of the issue.
--

REVIEWER	Roomaney, Rifqah South African Medical Research Council, Burden of Disease Research Unit, Tygerberg
REVIEW RETURNED	07-Mar-2022

GENERAL COMMENTS	Manuscript ID: bmjopen-2021-057464 Thank you for the opportunity to review the manuscript “Incidence of Hospitalized Acute Lower Respiratory Tract Disease Including Pneumonia and Lower Respiratory Tract Infection in Bristol, England, 2019”. The authors have done a good job on tackling an area of research that has been ignored in the UK for the past two decades. Abstract: • Page 3, Line 3: write out “aLRTD” and then give the acronym. Introduction: • The first sentence does not read well. Consider putting text in brackets: “e.g. including asthma 50 and chronic obstructive pulmonary disease (COPD), and acute heart failure (HF) events resulting in 51 respiratory symptoms (e.g. breathlessness).”• “Before the COVID-19 pandemic, European healthcare 52 costs for pneumonia alone were estimated at €10 billion annually,
---

	including €5.7 billion for inpatient 53 care. [1]” – Are these costs primarily associated with adults or children? Please clarify.  • Line 54 “in all studies pneumonia incidence increases sharply with age” – Can the authors clarify in the text whether these statistics exclude children. I assume it does. • Add reference/s to the sentence starting in Line 59. • Add reference/s to the sentence starting in Line 61. • Line 74: Heart failure was already defined so no need to define it again. • General comment: I think the authors could make a stronger case for why they chose to look at adults. Also, they need to discuss (either here or in the Discussion) how these estimates are helpful after the COVID-19 pandemic or how COVID would have affected the estimates. Methods:  • Line 95: If the records are electronic, the authors should state that and also state the name of the system used. • Can the authors reference a source to describe definitions of LRTI, pneumonia and Other LRTD? (Line 97 – 100). • Line 134: Does the GP practice registration data cover 100% of the population? Are there populations that may be excluded (e.g. new migrants?). Please consider adding a few lines on how the health system works in the UK as not all will be familiar. Results:  • Were all cases of aLRTD included? State if there were any records that could not be located? How many refusals? Discussion:  • General: Do the authors think there was a real increase in the aLRTD incidence? Or was it an effect of using a smaller denominator? If it was real, why would pneumonia be increasing in this population? • Another study limitation is that the data are not disaggregated by sex. The authors should review literature to state whether they expected a difference by sex or not. • Line 234 – 236. The consent process affects almost all studies. Authors should keep a record of how many refusals there were and state it (unless I’ve missed this). It’s difficult to assess if this was a factor without knowing how many people refused. • Again, I think some comment on how to interpret this data in light of COVID 19 would be useful. • Was it not possible to calculate the case fatality rate? Why not include this information if available?
--	---

REVIEWER	Llor, Carl Primary healthcare centre Barcelona-2B (Via Roma)
REVIEW RETURNED	08-Mar-2022

GENERAL COMMENTS	The study is methodologically sound but the main conclusions of this study are not original. It is already known that the presence of more acute lower respiratory tract disease, mainly pneumonia and other respiratory tract infections, is greater among the elderly and that the incidence of these conditions increases with age. I recognize, however, that the use of two methodologies, enrich the study and constitute the novel part of the study. The use of other codes, as clearly mentioned throughout the manuscript, is also a positive issue. Although the limitation section mentions how the results of this study can be extrapolated to other areas, even in the same
--

	country, they might be compromised, and in my opinion, this should be discussed more in depth. The prospective study was done in summer. Why did you choose this period when it's clearly known that the presence of lower respiratory tract infections is much lower? The links of the authors with Pfizer and the fact that a professional writer has written this study should be clarified in my opinion and discussed. Readership will ask why this study has been carried out. Why did you consider this definition of LRTS? You included heart failure, but you could have also included lung embolism or cardiac arrhythmia as they can cause breathlessness. You should discuss this and add a reference for your definition. Minor aspects: Considering pneumonia apart from lower respiratory tract infections is not fair. Despite only being mentioned in the main paragraph of the Discussion section, the correct term for the LRTI component is LRTI except pneumonia. I imagine you consider acute exacerbations of COPD and asthma as well as severe cases of acute bronchitis who need to be hospitalized as LRTI, but this should also be mentioned in the article. Why don't you show the number of each of these infections in the tables?
--	--

REVIEWER	Gong, Cynthia University of Southern California, Schaeffer Center for Health Policy & Economics
REVIEW RETURNED	09-Mar-2022

GENERAL COMMENTS	This paper provides important incidence information that has not been recently described, and should serve as a useful reference for other analyses related to acute LRTD / LRTIs in the future. The tables and figures are clear, and the methods are well-explained. I have only a few comments / questions for clarification listed below:  - Clinically, HF is not really considered an acute LRTD - can the authors provide a citation or further justification why the authors have classified heart failure, which is a cardiac condition, with pulmonary ones? The authors have provided some justification at the end of the introduction, but for clarity, it may help to explain upfront why heart failure is being considered aLRTD in this case - it is otherwise a bit misleading. - Why were the chosen ICD codes only considered within the first 5 positions? Were there ICD codes in further positions? - Can the authors comment on the true cost of LRTD given their findings? The intro cites a figure of €10 billion annually; do these findings change the authors' impression of that estimate?
---

VERSION 1 – AUTHOR RESPONSE

Reviewer: 1

Dr. D. Rebecca Prevots, National Institutes of Health

Comments to the Author:

1. An extremely thorough and well-presented analysis of the rate of Lower Respiratory Tract Infections across adult age groups in Bristol, England. The rates of LRTD were generally similar for both the retrospective and prospective components, which supports the validity of the findings. The low rates of microbiologic sampling are noteworthy, particularly in the older age groups. The strengths and limitations are clearly presented. Overall, this paper presents an informative and important review of the issue.

Thank you

Reviewer: 2

Ms. Rifqah Roomaney, South African Medical Research Council

Comments to the Author: Manuscript ID: bmjopen-2021-057464

Thank you for the opportunity to review the manuscript “Incidence of Hospitalized Acute Lower Respiratory Tract Disease Including Pneumonia and Lower Respiratory Tract Infection in Bristol, England, 2019”. The authors have done a good job on tackling an area of research that has been ignored in the UK for the past two decades.

Abstract:

1. Page 3, Line 3: write out “aLRTD” and then give the acronym.

Done – we have made this change. Page 3, line 3 now reads:

‘To determine the disease burden of acute lower respiratory tract disease (aLRTD) and its subsets...’

Introduction:

1. The first sentence does not read well. Consider putting text in brackets: “e.g. including asthma 50 and chronic obstructive pulmonary disease (COPD), and acute heart failure (HF) events resulting in 51 respiratory symptoms (e.g. breathlessness).”

Done. We have changed line 48-51 from:

Acute lower respiratory tract disease (aLRTD) encompasses pneumonia, lower respiratory tract infection (LRTI), acute bronchitis, exacerbation of underlying respiratory disease, (including asthma and chronic obstructive pulmonary disease (COPD), and acute heart failure (HF) events resulting in respiratory symptoms (e.g. breathlessness).

to:

Acute lower respiratory tract disease (aLRTD) encompasses pneumonia, lower respiratory tract infection (LRTI), acute bronchitis, exacerbation of underlying respiratory diseases (including asthma and chronic obstructive pulmonary disease [COPD]), and acute heart failure (HF) events resulting in respiratory symptoms (e.g. breathlessness).

1. “Before the COVID-19 pandemic, European healthcare costs for pneumonia alone were estimated at €10 billion annually, including €5.7 billion for inpatient care. [1]” – Are these costs primarily associated with adults or children? Please clarify.

Done. We have revised this to now read (line 51-52):

‘Before the COVID-19 pandemic, European healthcare costs for pneumonia alone in adults were estimated at €10 billion annually’

1. Line 54 “in all studies pneumonia incidence increases sharply with age” – Can the authors clarify in the text whether these statistics exclude children. I assume it does.

This does exclude children, and we have clarified the text in line 55 to now read:

‘however, in all studies pneumonia incidence in adults increases sharply with age’

1. Add reference/s to the sentence starting in Line 59.

Done

1. Add reference/s to the sentence starting in Line 61.

Done

1. Line 74: Heart failure was already defined so no need to define it again.

Done.

1. General comment: I think the authors could make a stronger case for why they chose to look at adults. Also, they need to discuss (either here or in the Discussion) how these estimates are helpful after the COVID-19 pandemic or how COVID would have affected the estimates.

Thank you. We chose to look at adults because of debate over accurate incidence in adults, including disease subsets within adults. By contrast, there are many studies of paediatric LRTI incidence. We have added to line 81-82:

‘There are many studies examining the incidence of acute respiratory illness in children; however, data on respiratory illness in adults in the UK is lacking.’

The study was conducted before the emergence of COVID-19. This new respiratory pathogen in and of itself will have changed the epidemiology of acute LRTD; additionally, public health measures such as social distancing, mask-wearing, vaccination programs etc will also have considerably changed the epidemiology and consequently incidence of aLRTD. Therefore disease incidence may have risen (due to COVID in addition to other aLRTD), stayed the same (for example if COVID replaced other aLRTD) or fallen (in which despite the emergence of COVID the public health measures reduced other aLRTD such that the sum was less overall). Whilst it is not possible to be certain which of these will occur in the context of COVID-19, we have added (line 304-307):

‘This study was conducted before the emergence of COVID-19, and we think these data will be useful in one of two ways in the context of COVID-19: (1) either COVID-19 will become endemic, and the data will reflect the first year before a new normal, or (2) COVID-19 will abate and it will provide an anchor for understanding incidence during a respiratory viral pandemic’

Methods:

1. Line 95: If the records are electronic, the authors should state that and also state the name of the system used.

Done. Line 99-100 now reads:

'For the retrospective analysis, all adult inpatient admissions (≥ 18 years) obtained from Hospital Episode Statistic (HES) to the study hospital'

1. Can the authors reference a source to describe definitions of LRTI, pneumonia and Other LRTD? (Line 97 – 100).

Done. In the interest of brevity, we have added this to the Supplementary Data 2.

1. Line 134: Does the GP practice registration data cover 100% of the population? Are there populations that may be excluded (e.g. new migrants?). Please consider adding a few lines on how the health system works in the UK as not all will be familiar.

In the UK healthcare system, any one is entitled to register with a GP for no charge – regardless of residential status. This includes new migrants and people who live in the UK without visas (ie illegal immigrants) as the NHS GP system provides universal healthcare free at the point of access.

We have therefore added to line 144-145:

'In the UK, GP registration is available free of charge for all, regardless of residential status'

Results:

1. Were all cases of aLRTD included? State if there were any records that could not be located? How many refusals?

Within the prospective audit, all hospitalisations were reviewed and there are therefore no missing records or cases. There were no refusals for use of data. In the retrospective dataset, all data was obtained from HES (hospital episode statistics) available at the hospital where the study was undertaken. Within this dataset, there were no lines of missing data and so there are no missing data. However, we do discuss the limitations of using ICD-10 coding data and are aware of this limitation of the study (line 289-299).

We have also clarified this on line 116-117, by adding:

'There were no patient refusals for either approach.'

Discussion:

1. General: Do the authors think there was a real increase in the aLRTD incidence? Or was it an effect of using a smaller denominator? If it was real, why would pneumonia be increasing in this population?

Thank you. As this study is the first to comprehensively assess aLRTD incidence in hospitalised adults in the UK, and given the reasons why disease incidence may be underestimated it is difficult to be certain whether the increase in incidence we see is a 'real increase', a result of more accurate recording, or a combination of these two factors.

We feel that line 200 states our position that:

'These results suggest rates are probably significantly higher than previous disease estimates from the UK (Table 4) but comparable with many results globally'

The authors feel that this is an important question to address, but that it cannot be addressed by the data within this study in comparison to others due to the methodological differences, and this is why additional research is required to ensure accurate capture of all aLRTD events. We have therefore added to line 310:

'and these estimates were higher than those obtained previously in the UK. Determining if there is a real increase in incidence, or if this estimate is larger due to more accurate methodology including a more accurate denominator will require ongoing comprehensive surveillance'

1. Another study limitation is that the data are not disaggregated by sex. The authors should review literature to state whether they expected a difference by sex or not.

The retrospective analysis (Table 1, page 19) shows total disease and the pneumonia, NP-LRTI, HF and other aLRTD subgroups broken into sex.

1. Line 234 – 236. The consent process affects almost all studies. Authors should keep a record of how many refusals there were and state it (unless I've missed this). It's difficult to assess if this was a factor without knowing how many people refused.

This study did not require consent as both the retrospective and prospective data were obtained as part of an audit, and are available to allow for the planning of hospital services and patient care. No individuals within the cohort utilised the national data-opt out and as such, there were therefore no refusals in this study.

*We have also already addressed this in line 116-117 by adding:
'There were no patient refusals for either approach.'*

1. Again, I think some comment on how to interpret this data in light of COVID 19 would be useful.

Thank you, we agree that the emergence of COVID-19 has changed the epidemiology of acute respiratory disease, but the authors prefer not to speculate without further data and would like to reserve this for a future publication. We have, however added on line 304-307:

'This study was conducted before the emergence of COVID-19, and we think these data will be useful in one of two ways in the context of COVID-19: (1) either COVID-19 will become endemic and the data will reflect the first year before a new normal, or (2) COVID-19 will abate and it will provide an anchor for understanding incidence during a respiratory viral pandemic'

1. Was it not possible to calculate the case fatality rate? Why not include this information of available?

Unfortunately this was not possible with the data available for this study.

Reviewer: 3

Dr. Carl Llor, Primary healthcare centre Barcelona-2B (Via Roma)

Comments to the Author:

The study is methodologically sound but the main conclusions of this study are not original. It is already known that the presence of more acute lower respiratory tract disease, mainly pneumonia and other respiratory tract infections, is greater among the elderly and that the incidence of these conditions increases with age. I recognize, however, that the use of two methodologies, enrich the study and constitute the novel part of the study. The use of other codes, as clearly mentioned throughout the manuscript, is also a positive issue.

1. Although the limitation section mentions how the results of this study can be extrapolated to other areas, even in the same country, they might be compromised, and in my opinion, this should be discussed more in depth.

Done - we agree with this comment, and have strengthened the discussion around the study limitations. Line 283-293 was previously:

'However, the study also had some limitations. This was a single-center study, with a predominantly Caucasian cohort; therefore, the findings might not be generalizable to other populations. The ICD-10 coding data analysis was limited to codes within the first five positions, and therefore may have excluded some cases where other diagnoses were placed higher in the diagnostic coding hierarchy. Furthermore, we could not determine how many cases of the 28.1% ICD-10 cases also coded with nosocomial infection had hospital-acquired respiratory infection rather than other nosocomial infections.'

and now reads:

'However, the study also had some limitations. This was a single-center study, with a predominantly Caucasian cohort; therefore, the findings might not be generalizable to other populations both within the UK and in other countries. Different healthcare systems may affect patient treatment preference, and as the NHS provides care which is free at the point of access, the hospitalization rates seen in this study may be different in fee or insurance based healthcare system. Similarly, physician treatment preferences may affect hospitalization rates, and we have not explored these in this analysis. The ICD-10 coding data analysis was limited to codes within the first five positions, and therefore may have excluded some cases where other diagnoses were placed higher in the diagnostic coding hierarchy. Furthermore, we could not determine how many cases of the 28.1% ICD-10 cases also coded with nosocomial infection had hospital-acquired respiratory infection rather than other nosocomial infections.'

1. The prospective study was done in summer. Why did you choose this period when it's clearly known that the presence of lower respiratory tract infections is much lower?

Thank you. This prospective work was undertaken in August to September, which in the UK is not the summer and represents the start of the 'winter respiratory infection season'. We selected these months because they were considered to represent average admissions from acute respiratory disease (as shown in Figure 1), and therefore admission numbers would not be subject to excess respiratory cases in winter nor reduced cases in the summer.

We have added to line 109-111:

'This time period was selected because it was felt to represent a period when there were an average number of adults hospitalised with aLRTD.'

1. The links of the authors with Pfizer and the fact that a professional writer has written this study should be clarified in my opinion and discussed. Readership will ask why this study has been carried out.

The study was carried out to meet the study objectives, which are stated on line 82-85:

'Given the paucity of data supporting accurate aLRTD incidence rates and its disease subsets in adults, we undertook to assess aLRTD incidence by two approaches (retrospective and prospective) in Bristol, UK, seeking to determine the disease burden of hospitalized aLRTD and its subgroups more accurately.'

We have declared that a professional writer was used, and that they undertook the literature review on behalf of the authors. The manuscript was prepared by the authors and the medical writer performed the referencing. The authors have also completed a conflict-of-interest declaration for all authors, including those who are employees of Pfizer (page 2). This meets both ICJME authorship and contributorship reporting recommendations, and those for this publishing house and journal.

1. Why did you consider this definition of LRTD? You included heart failure, but you could have also included lung embolism or cardiac arrhythmia as they can cause breathlessness. You should discuss this and add a reference for your definition.

Thank you. We feel this is covered in our response to Reviewer 4 (point 25) and Reviewer 2 (point 11)

Minor aspects:

1. Considering pneumonia apart from lower respiratory tract infections is not fair. Despite only being mentioned in the main paragraph of the Discussion section, the correct term for the LRTI component is LRTI except pneumonia.

We agree with the reviewer that respiratory infection encompasses pneumonia and non-pneumonic respiratory infection (NP-LRTI), and to not include NP-LRTI would result in an undercalculation of the total burden of respiratory infection – and would like to thank the reviewer for bringing this to our attention. We have amended the nomenclature throughout the manuscript, so that where we previously referred to 'LRTI' we now use the term 'NP-LRTI' and have defined this in line 49.

1. I imagine you consider acute exacerbations of COPD and asthma as well as severe cases of acute bronchitis who need to be hospitalized as LRTI, but this should also be mentioned in the article. Why don't you show the number of each of these infections in the tables?

We agree with the reviewer the chronic respiratory disease causes significant disease burden and hospitalisations, and that it is important to accurately define incidence to provide more appropriate care for such patients. On line 48, we defined aLRTD as including exacerbation of chronic respiratory disease – and this would encompass patients with asthma and COPD. Exacerbation can be either infective (e.g. caused by pneumonia), non-infective (e.g. caused by a pneumothorax), or both (e.g. when a patient with chronic respiratory disease has both an infection and non-infective cause of their respiratory deterioration, such as a pneumomediastinum). Additionally, there may or may not be an element of acute cardiac failure that causes acute respiratory disease in patients with chronic lung disease.

Elucidating the precise overlap between chronic respiratory disease exacerbation, heart failure, respiratory infection and non-infective causes of any individual exacerbation is often complex and there is frequently clinical uncertainty. We therefore made the pragmatic decision not to explore this complex interaction within the data set that was available for this manuscript, and to collect

additional data for a future publication to highlight the importance of this under-appreciated disease burden.

Reviewer: 4

Dr. Cynthia Gong, University of Southern California Comments to the Author:

This paper provides important incidence information that has not been recently described, and should serve as a useful reference for other analyses related to acute LRTD / LRTIs in the future. The tables and figures are clear, and the methods are well-explained. I have only a few comments / questions for clarification listed below:

1. Clinically, HF is not really considered an acute LRTD - can the authors provide a citation or further justification why the authors have classified heart failure, which is a cardiac condition, with pulmonary ones? The authors have provided some justification at the end of the introduction, but for clarity, it may help to explain upfront why heart failure is being considered aLRTD in this case - it is otherwise a bit misleading.

Thank you. We considered cardiac failure with respiratory symptoms as being acute LRTD because there is substantial evidence that respiratory viral infection, particularly RSV, can cause either acute heart failure or an episode of heart failure 3-4 weeks after the primary infection. Patients admitted with cardiac failure are not routinely tested for respiratory infection, however work from Ann Falsey and others suggest that this is an under-recognised clinical phenomenon and we therefore wished to ensure this was captured within the data presented here. We have therefore added to line 57-59:

'Whilst HF is not typically clinically included as an acute respiratory illness, HF with respiratory symptoms may be caused by respiratory viral infection, such as respiratory syncytial virus (RSV), either acutely or 3-4 weeks after the primary infection'

26. Why were the chosen ICD codes only considered within the first 5 positions? Were there ICD codes in further positions?

The authors only had access to the ICD-10 codes within the first 5 positions, and beyond that the data was unfortunately not available to us. We have added to line 105:

'Only the first 5 ICD-10 codes were available for analysis.'

1. Can the authors comment on the true cost of LRTD given their findings? The intro cites a figure of €10 billion annually; do these findings change the authors' impression of that estimate?

Thank you. The NHS system does not attach a cost per patient episode for each hospitalisation, and it is therefore unfortunately not possible for us to determine the cost of aLRTD with the data, and we regret that this is beyond the scope of this manuscript.

VERSION 2 – REVIEW

REVIEWER	Roomaney, Rifqah South African Medical Research Council, Burden of Disease Research Unit, Tygerberg
REVIEW RETURNED	30-Mar-2022

GENERAL COMMENTS	All issues have been addressed. Thank you for the opportunity to review the manuscript.
REVIEWER	Llor, Carl Primary healthcare centre Barcelona-2B (Via Roma)
REVIEW RETURNED	30-Mar-2022
GENERAL COMMENTS	Queries appropriately addressed.